# Abundance, size, and survival of recruits of the reef coral *Pocillopora acuta* under ocean warming and acidification

**Keisha D. Bahr**[1]*, Tiana Tran[2], Christopher P. Jury[2], Robert J. Toonen[2]

**1** Department of Life Sciences, Texas A&M University-Corpus Christi, Corpus Christi, Texas, United States of America, **2** Hawai'i Institute of Marine Biology, University of Hawai'i at Mānoa, Kāne'ohe, Hawaii, United States of America

* Keisha.Bahr@tamucc.edu

## Abstract

Ocean warming and acidification are among the greatest threats to coral reefs. Massive coral bleaching events are becoming increasingly common and are predicted to be more severe and frequent in the near future, putting corals reefs in danger of ecological collapse. This study quantified the abundance, size, and survival of the coral *Pocillopora acuta* under future projections of ocean warming and acidification. Flow-through mesocosms were exposed to current and future projections of ocean warming and acidification in a factorial design for 22 months. Neither ocean warming or acidification, nor their combination, influenced the size or abundance of *P. acuta* recruits, but heating impacted subsequent health and survival of the recruits. During annual maximum temperatures, coral recruits in heated tanks experienced higher levels of bleaching and subsequent mortality. Results of this study indicate that *P. acuta* is able to recruit under projected levels of ocean warming and acidification but are susceptible to bleaching and mortality during the warmest months.

## Introduction

Coral reefs are one of the most diverse and productive ecosystems in the world, but are undergoing ecological decline due to the increasing diversity, frequency, and scale of human impacts [1]. Increased fossil fuel burning has forced the atmospheric carbon dioxide ($CO_2$) concentration to increase at a severe rate, intensifying the greenhouse effect and warming the planet and its oceans. The rate of current $CO_2$ increase is exponentially (dependent on emission scenarios 200-600x) faster than what has occurred over the past 450,000 years, and the rising atmospheric $CO_2$ levels have risen to the point of being irreversible on human timescales [2–4]. These increases in $CO_2$ levels cause the planet to warm and the oceans to acidify, which has severe implications on corals. Corals occupy a narrow thermal range, becoming stressed if their thermal threshold is exceeded, causing them to bleach (i.e., loss of symbiotic algae or algal pigments) [4]. Corals throughout the world are living within 1–2°C of their upper thermal limit during the summer months in both tropical and subtropical environments [5]. Almost every coral reef region in the world has now suffered extensive stress and/or mortality

**Funding:** This paper is funded in part by the National Science Foundation (OA no. 1416889), and by the National Oceanic and Atmospheric Administration, Project R/IR-23, R/IR-32, which is sponsored by the University of Hawaii Sea Grant College Program, SOEST, under Institutional grant no. NA09OAR4170060, NA14OAR4170071 from NOAA Office of Sea Grant, Department of Commerce. The funders had no role in study design, data collection and analysis, decision to publish, or preparation of the manuscript.

**Competing interests:** The authors have declared that no competing interests exist.

associated with increases in global sea surface temperatures (SSTs) over the past century due to anthropogenic emissions [6]. With projected increases in ocean temperatures, corals are expected to frequently exceed their upper lethal limits by 2030 [4, 7, 8] in the absence of rapid adaptation [9].

Nearly one fourth of $CO_2$ emitted from all anthropogenic sources enters the ocean, increasing acidity and decreasing the pH of the ocean [4]. Ocean acidity has increased by approximately 25% (decrease of 0.1pH units) since pre-industrial times [10] and further acidification is expected to result in severe negative effects on calcifying marine organisms [11–16]. Calcification rates of reef building corals are estimated to decrease up to 15–20% by the end of the century due to ocean acidification alone [3, 14, 17–25] although, field studies have revealed that abiotic and biotic factors in the marine environment may mask the effect of ocean acidification on corals, obscuring its detection [26–28], and biological feedbacks may complicate such predictions [29]. With observed increases in the frequency and severity of massive coral bleaching events alongside increases in ocean acidity, many predict that reefs may collapse within the next few decades [4, 14].

Coral reef recovery from disturbance, and persistence in the face of environmental change, is dependent on both the rate of adult mortality and the ability of recruitment to replace individuals lost from the population [30, 31]. Coral recruitment requires production of viable larvae, survival of those offspring through successful settlement and post-settlement growth until individuals join the local breeding population [32]. Early life stages of corals have been shown to be particularly sensitive to changes in temperature and pH levels [33–38]. For example, future projections of ocean acidification may inhibit skeletal growth of newly settled coral recruits by 20–60% [39–41]. Successful recruitment and high rates of juvenile calcification are critical to maintaining coral reef health and ultimately the viability of coral reef ecosystems [42]. Therefore, it is critical to examine how future climate change conditions will affect coral recruitment, growth, and survivorship in the early stages of development, when recruits are especially vulnerable to mortality (e.g., due to mechanical damage, overgrowth, predation, etc.) to better predict future trajectories of coral reef ecosystems.

Continuous flow-through experimental mesocosms were held at current and future projections of ocean warming and acidification to understand long-term impacts on coral reef communities. During this long-term experiment, we noticed high levels of recruitment within these experimental mesocosms. This provided a fortuitous opportunity to understand the impacts of ocean warming and acidification on coral recruitment (i.e., abundance, size, and survival) of *Pocillopora acuta* that recruited inside these flow-through experimental mesocosms. Based on prior results, we hypothesized that corals recruits inside acidified and/or warmed conditions would be lower in number, smaller in size, and would experience higher levels of mortality as compared to present-day control conditions.

## Methods

### Experimental setup

This study was conducted in a flow-through mesocosm facility at the Hawaiʻi Institute of Marine Biology on Moku o Loʻe in Kāneʻohe Bay, Hawaiʻi. The 40-tank (70 L, 50 x 50 x 30 cm) outdoor system was exposed to a fully factorial design that consisted of two levels each of temperature (present-day vs. high) and $pCO_2$ (present-day vs. high), resulting in four treatments total (n = 10 mesocosms per treatment) for 22 months. Unfiltered seawater was pumped directly from the adjacent coral reef, which allowed for natural seasonal and diel fluctuations in environmental parameters in the mesocosms. The incoming seawater was directed to a large sump where temperature and chemistry were adjusted to average present-day conditions,

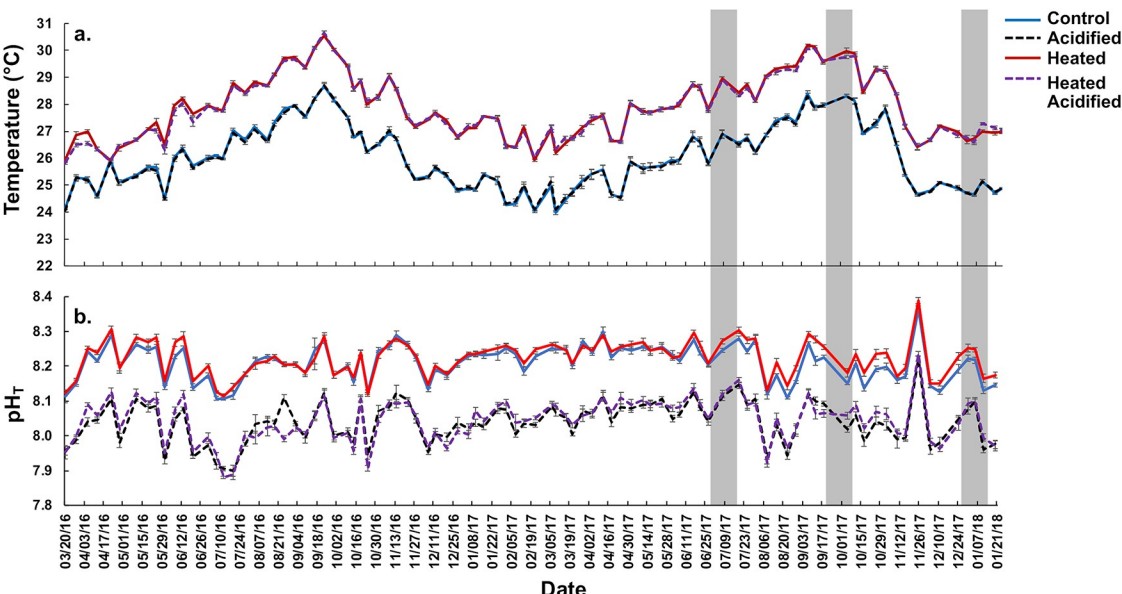

**Fig 1. Mean mid-day (~12:00 h) temperatures (°C) (a.) and pH$_T$ (b.) in experimental treatments of control (AT ACO2; blue solid line), acidified (AT HCO2; black dotted line), heated (HT ACO2; red solid line), and heated acidified (HT HCO2; purple dotted line) throughout the 22-month experimental period.** Survey periods (i.e., July 2017, October 2017, and January 2018) are shaded. Error bars represented SE of the mean (n = 10).

before being pumped into the header tanks where additional heat and pCO$_2$ manipulations were made according to treatment. Temperature of the incoming water was adjusted to target values using a commercial heat pump (Titan HP 53) with a temperature controller to provide cooling or heating as needed. Adjustments to pCO$_2$ were achieved by using high-precision needle valves connected to seawater pumps to mix a precise quantity of CO$_2$ gas into the seawater [43]. Heating was provided by commercial aquarium heaters on temperature controllers (1800 watt titanium heaters manufactured by Process Technology). The control treatment was adjusted to the average present-day offshore temperature and pCO$_2$ conditions, which is ~23.5–27.5°C over the annual cycle and ~400μatm pCO$_2$ (Fig 1). The heated treatments were ~2°C greater than the control, and the acidified treatments were ~0.2 pH units lower (~350μatm pCO$_2$ greater), corresponding to expected future conditions under a moderately high CO$_2$ emissions scenario as outlined by the International Panel on Climate Change [44].

These experimental mesocosms were a part of a larger long-term project to understand ocean warming and acidification impacts on coral reef communities. To simulate a reef environment, each mesocosm included coral nubbins from all the major Hawaiian reef building species (*Montipora capitata*, *Montipora flabellata*, *Montipora patula*, *Pocillopora acuta*, *Pocillopora meandrina*, *Porites compressa*, *Porites evermanni*, and *Porites lobata*), yielding an initial coral cover of roughly 10%, a thin layer of carbonate reef sand, and pieces of reef rubble. A juvenile threadfin butterflyfish (*Chaetodon auriga*) and a juvenile convict tang (*Acanthurus triostegas*) were placed in each mesocosm to graze algae and non-coral invertebrates and provided the essential ecological functions of herbivory and predation in the mesocosms (S1 Fig). Mesocosms were not cleaned and diverse assemblages of algae and invertebrates recruited into them during the experiment. All *P. acuta* which recruited into the tanks were assessed in this study. The vast majority of which settled on the mesocosm walls, though a small number occurred on other surfaces (e.g., drain pipe, bottom grate, etc.).

## Experimental treatments

Four experimental treatments were maintained for this study: control (AT ACO2), acidified only (AT HCO2), heated only (HT ACO2), and dual stress heated acidified (HT HCO2). Weekly mid-day (~1200 h) water parameters [salinity (psu), and temperature (˚C)] were measured using a YSI multiparameter meter (YSI 556MPS, accuracy ± 0.15˚C ± 0.2mg l$^{-1}$, ±2 $^o/_{oo}$). Water parameters (temperature, salinity, dissolved oxygen, $pH_T$) were measured to confirm desired environmental and chemical conditions (Table 1). Measurements of $pH_T$ were assessed spectrophotometrically using *m*-cresol purple dye (Sigma-Aldrich$^©$ #857890) according to SOP 7 [45]. Total alkalinity was measured using an automatic titrator (Titrino$^©$ Plus 877, Metrohm) with pH glass electrode (9101 Herisau, Metrohm$^©$). Precision of the automatic titrator was confirmed with certified reference materials (CRM) (from A. Dickson Laboratory, Scripps Institution of Oceanography). Temperature, total alkalinity, $pH_T$ and salinity were used to calculate $pCO_2$ in CO2SYS [46] with the stoichiometric dissociation constants (K1, K2) defined by Mehrbach et al. [47] and refit by Dickson and Millero [48].

## Experimental species

A small number of corals were first observed recruiting along the walls of the experimental mesocosms in June 2016, after about 3 months from the start of the experiment, and recruitment appeared to cease after October in 2016. A substantially larger number of coral recruits began to appear in the mesocosms in June 2017 and recruitment appeared to end after October 2017, as in the previous year. These corals were identified as *Pocillopora acuta* recuits [49]. *Pocillopora acuta* has been observed to sexually spawn on the first quarter moon and four days after full moon [49, 50] and potentially has mixed mode of reproduction as observed in its sister species *Pocillopora damicornis* releasing brooded asexual larvae after new moon [51]. Like *P. damicornis*, *P. acuta* tends to grow quickly, mature early, and brood larvae that are ready to settle within as little as hours to a couple of days after release [52]. This short pelagic larval duration facilitated the recruitment of *P. acuta* on the sides of the mesocosms in high numbers, which presented an opportunity to examine the effects of elevated temperature and acidification on recruitment, growth, and survival of *P. acuta* in a long-term mesocosm setup.

**Table 1. Monthly mid-day mean environmental conditions for experimental treatments during coral recruit assessment periods in July 2017, October 2017, and January 2018.**

| Sampling Month | Treatment | Salinity (psu) | | Temperature (˚C) | | Total Alkalinity μmol kg$^{-1}$ | | $pH_T$ | | $pCO_2$ μatm | |
|---|---|---|---|---|---|---|---|---|---|---|---|
| | | Mean | ± SE | Mean | ± SE | Mean | ± SE | Mean | ± SE | Mean | ± SE |
| July 2017 | AT ACO2 | 34.65 | 0.01 | 26.61 | 0.07 | 2063.03 | 9.81 | 8.21 | 0.01 | 218.74 | 6.91 |
| | AT HCO2 | 34.65 | 0.01 | 26.58 | 0.06 | 2064.58 | 9.67 | 8.06 | 0.01 | 354.67 | 15.80 |
| | HT ACO2 | 34.67 | 0.01 | 28.57 | 0.05 | 2085.78 | 8.82 | 8.20 | 0.01 | 227.23 | 7.90 |
| | HT HCO2 | 34.67 | 0.01 | 28.49 | 0.05 | 2080.57 | 8.42 | 8.03 | 0.01 | 387.84 | 18.49 |
| October 2017 | AT ACO2 | 34.65 | 0.02 | 27.65 | 0.09 | 2088.10 | 7.92 | 8.11 | 0.01 | 297.80 | 8.36 |
| | AT HCO2 | 34.65 | 0.02 | 27.68 | 0.09 | 2086.59 | 7.61 | 7.95 | 0.01 | 477.89 | 19.67 |
| | HT ACO2 | 34.67 | 0.02 | 29.41 | 0.10 | 2119.56 | 6.50 | 8.12 | 0.01 | 300.78 | 11.01 |
| | HT HCO2 | 34.68 | 0.02 | 29.34 | 0.08 | 2121.77 | 5.64 | 7.96 | 0.01 | 476.09 | 20.60 |
| January 2018 | AT ACO2 | 34.47 | 0.01 | 24.84 | 0.04 | 2207.23 | 12.00 | 8.20 | 0.02 | 253.04 | 10.42 |
| | AT HCO2 | 34.48 | 0.01 | 24.84 | 0.04 | 2209.05 | 12.02 | 8.03 | 0.02 | 417.41 | 23.14 |
| | HT ACO2 | 34.49 | 0.01 | 26.91 | 0.05 | 2215.73 | 11.40 | 8.19 | 0.02 | 257.13 | 11.42 |
| | HT HCO2 | 34.49 | 0.01 | 27.01 | 0.05 | 2218.55 | 10.96 | 8.01 | 0.02 | 444.70 | 24.63 |

## Coral recruit assessment: Health, size, and abundance

Recruits of *Pocillopora acuta* were sized, counted, and their condition was assessed within each of the forty tanks in July 2017, October 2017, and January 2018. The maximum diameter for each recruit was measured using a caliper to the nearest 1 mm. Abundance was determined at the end of each assessment by totaling the number of individual corals found in each mesocosm. Recruit health was assess through visual assessments of pigmentation, which was denoted as normal, pale, bleached, dead, or a combination of more than one expressed as a percentage [53]. Observer error in coral health assessment was reduced by standardizing to the same observer taking the measurements throughout all assessments (i.e., July, October and January).

Coral recruit measurements and assessments were completed within a one-week time frame to minimize the possibility of changes to the corals' conditions among mesocosms over time.

Pre-bleaching assessment was conducted 5–10 July, 2017. The bleaching assessment was conducted 12–17 October, 2017 following the peak of high temperature. The final recovery assessment was conducted in 18–24 January 2018 (Fig 1).

## Statistical analysis

**Environmental data.** Mid-day (~1200 h) environmental conditions [temperature (˚C), salinity (psu), total alkalinity (µmol kg$^{-1}$), pH$_T$, and pCO$_2$ (µatm)] were averaged acrossed sampling month (July 2017; October 2017; and January 2018) (Table 1). Differences between temperatures and pH among treatments was analyzed using a One-way ANOVA.

**Coral recruit abundance.** The total number of recruits per treatment is presented, as is the average number per tank. The abundance of *P. acuta* recruits per survey period (i.e., July, October, and January), for each treatment (i.e., AT ACO2; AT HCO2; HT ACO2; HT HCO2), were compared by a 2 × 2 contingency table and a Pearson's Chi-squared test.

**Coral recruit size.** Initial size (July assessment only) of coral recruits (mean within each tank) by treatment was assessed using a Kruskall Wallis test. A match paired t-test was used to determine the change in mean recruit size within each tank across the assessment periods (July 2017, October 2017, and January 2018) as a proxy for growth rates.

**Coral recruit health.** Pigmentation was denoted as normal, pale, bleached, dead, or a combination of more than one expressed as a percentage (e.g., 50% dead and 50% bleached). Due to unequal variance, mean percent health (i.e., normal, pale, bleached, dead) by treatment was analyzed using a Kruskall Wallis test.

# Results

## Environmental conditions

Seawater physical and chemical properties varied naturally throughout the 22-month experiment over diurnal and seasonal cycles. Among the sampling months (Jul 2017, Oct 2017, and Jan 2018), the warmest temperatures were observed in October 2017 in control tanks (~27.6 ± 0.1) and heated tanks (~29.4 ± 0.1). The coolest temperatures were observed in January 2018 in control tanks (~24.8 ± 0.1) and heated tanks (~27.0 ± 0.1). Salinity remained relatively constant during the experiment with slightly lower levels in January 2018 (34.5 psu) (Table 1). Heated treatments maintained similar temperatures throughout the experiment (p>0.05), but significantly higher than control treatments (One-way ANOVA; F$_{(3,464)}$ = 101.9; p<0.0001). Similarly, pH$_T$ was similar among acidified treatments (p>0.05) but significantly

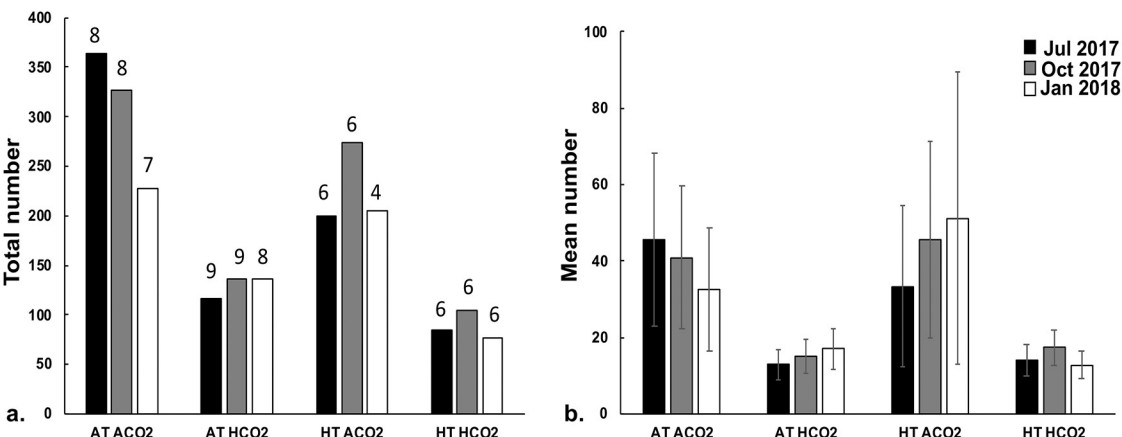

**Fig 2. Total (a.) and mean (b.) number of *Pocillopora acuta* recruits within each treatment: Control (AT ACO2), acidified (AT, HCO2), heated (HT, ACO2), and heated acidified (HT, HCO2) pooled acrossd tanks in July (black), October (grey), and January (white).** Numbers above bar indicate number of tanks coral recruits were found in (total number of tanks per treatment is 10). Error bars are SE of the mean.

lower than control (One-way ANOVA; $F_{(3,464)}$ = 400.9; p<0.0001). Diel sampling revealed the highest temperature and $pH_T$ at mid-day (12:00) within all treatments (S2 Fig).

## Coral recruit abundance

The total number of *Pocillopora acuta* recruits varied within and between treatments (Fig 2). In the heated treatments (i.e., HT ACO2 and HT HCO2), only six tanks (out of 10 replicated tanks) had at least 1 coral recruit present during the July assessment. The highest number of tanks with *P. acuta* recruits was in the acidified (AT HCO2) only tanks (in 9 out of the 10 replicated tanks). This pattern held during the following assessments (Fig 2).

The total number of individuals across treatments did not vary in July (Pearson's Chi-square; df = 1; N = 765; p = 0.0856) or October (Pearson's Chi-square; df = 1; N = 841; p = 0.5523). Heating and acidification alone and in combination did not have a significant impact on the number of recruits during the July and October surveys; however, counts were significantly lower than expected in the heated tanks as compared to the control temperature tanks in the January 2018 assessment (Pearson's Chi-square; df = 1; N = 646; p = 0.0070) (Table 2).

## Recruit size

Initial assessment (July 2017) of recruit size did not vary across treatments (Kruskal Wallis; df = 3 p = 0.2185). Comparison of mean coral recruit size across assessment periods (July to

**Table 2. Chi-square analysis of the number of *P. acuta* recruits across treatments during assessments conducted in July 2017 (pre-bleaching), October 2017 (bleaching), and January 2018 (recovery).**

| Jul-17 | | Temperature | | | Oct-17 | | Temperature | | | Jan-18 | | Temperature | | |
|---|---|---|---|---|---|---|---|---|---|---|---|---|---|---|
| | | Control | Heated | Total | | | Control | Heated | Total | | | Control | Heated | Total |
| CO2 | Control | 364 | 200 | 564 | CO2 | Control | 327 | 274 | 601 | CO2 | Control | 228 | 205 | 433 |
| | *expected* | *354* | *210* | | | *expected* | *331* | *270* | | | *expected* | *244* | *189* | |
| | Acidified | 116 | 85 | 201 | | Acidified | 136 | 104 | 240 | | Acidified | 136 | 77 | 213 |
| | *expected* | *126* | *75* | | | *expected* | *132* | *108* | | | *expected* | *120* | *93* | |
| | Total | 480 | 285 | **765** | | Total | 463 | 378 | **841** | | Total | 364 | 282 | **646** |

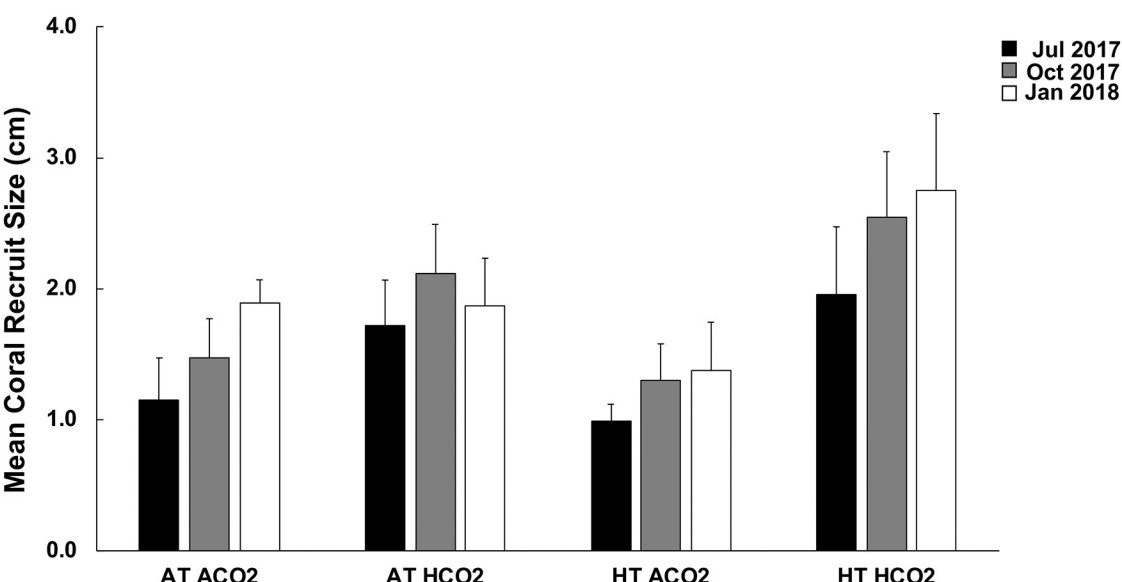

**Fig 3. Mean maximum diameter *Pocillopora acuta* recruit size (cm) pooled across tank means within treatments of control (AT ACO2, *n = 8*), acidified (AT, HCO2, *n = 9*), heated (HT, ACO2, *n = 6*), and heated acidified (HT, HCO2, *n = 6*) during initial (July 2017, black), bleaching (October 2017, grey), and recovery assessment (January 2018, white).** Error bars are SE of the mean.

October) reveal significant increases (mean difference ± SE) in size of recruits in the control (0.32 ± 0.09 cm; p = 0.0049), acidified (0.40 ± 0.17 cm; p = 0.0226), and heated acidified conditions (0.60 ± 0.13 cm; p = 0.0027). Size of coral recruits in the heated conditions remained the same between the July and October assessments (matched pairs; p = 0.1425) (Fig 3). No significant difference in size was observed between October 2017 and January 2018 in any of the treatments (control p = 0.0728; acidified p = 0.7925; heated 0.2748; heated acidified 0.1368) (Fig 4).

## Coral health assessment

Initial assessment (July 2017) of coral recruit health revealed high levels of normal pigmentation with moderate paling that was similar across treatments (Kruskal Wallis; Normal p = 0.1670; Pale p = 0.1224). During the October assessment, which was conducted following the peak of the annual high temperature, high levels of bleaching (49–56%) and mortality (4–7%) were observed in recruits in the heated tanks (Fig 5). These levels were significantly higher than coral recruits observed in the non-heated tanks (Kruskal Wallis; Bleaching p = 0.0006;

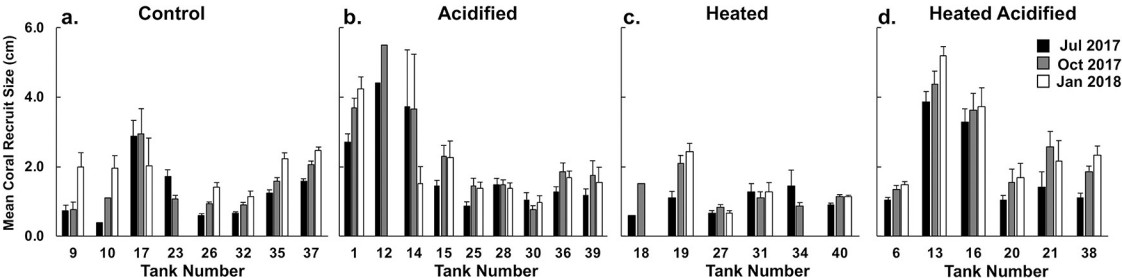

**Fig 4. Mean maximum diameter *Pocillopora acuta* recruit size (cm) by tank in control (a.), acidified (b.), heated (c.) and heated acidified (d.) conditions during the July (black), October (grey), and January (white) assessments.**

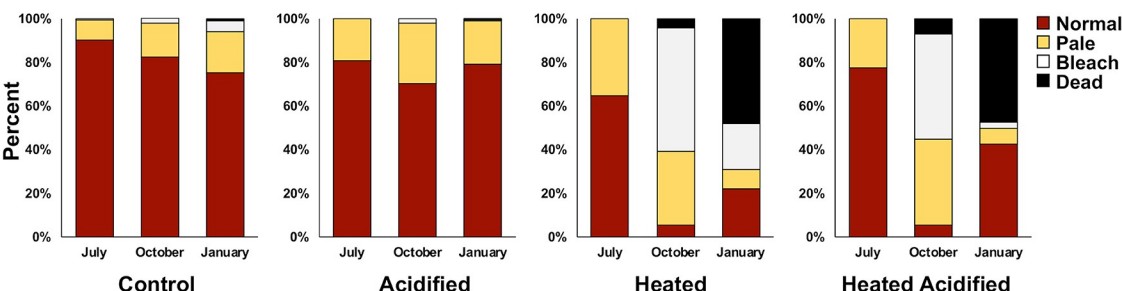

**Fig 5. Mean coral recruit health assessment of each treatment (pooled across tanks) (control, acidified, heated, heated acidified) during each survey (July 2017, October 2017, and January 2018).** Coral health assessment is denoted by pigmentation of normal (red), pale (yellow), bleached (white), and dead (black).

**Table 3. Mean coral recruit health by assessment period (July 2017, October 2017, and January 2018) in control, acidified, heated, and heated acidified treatments.**

| Mean Response (%) | | Control | Acidified | Heated | Heated Acidified |
|---|---|---|---|---|---|
| | | Mean ± *SE* | Mean ± *SE* | Mean ± *SE* | Mean ± *SE* |
| Jul-17 | Normal | 89.99 ± 3.36 | 80.81 ± 7.02 | 64.68 ± 14.67 | 77.62 ± 4.38 |
| | Pale | 9.44 ± 3.09 | 19.19 ± 7.02 | 35.32 ± 14.67 | 22.38 ± 4.38 |
| | Bleached | 0.54 ± 0.54 | 0.00 ± 0.00 | 0.00 ± 0.00 | 0.00 ± 0.00 |
| | Dead | 0.00 ± 0.00 | 0.00 ± 0.00 | 0.00 ± 0.00 | 0.00 ± 0.00 |
| Oct-17 | Normal | 82.89 ± 2.49 | 71.10 ± 9.16 | 5.25 ± 3.30 | 5.68 ± 3.63 |
| | Pale | 15.13 ± 1.91 | 27.31 ± 9.25 | 34.05 ± 9.68 | 40.52 ± 13.15 |
| | Bleached | 2.17 ± 1.26 | 2.23 ± 1.12 | 56.33 ± 9.62 | 49.17 ± 11.33 |
| | Dead | 0.00 ± 0.00 | 0.00 ± 0.00 | 4.43 ± 2.39 | 7.14 ± 6.43 |
| Jan-18 | Normal | 75.20 ± 6.11 | 78.77 ± 3.74 | 21.99 ± 13.93 | 42.32 ± 13.92 |
| | Pale | 19.01 ± 5.58 | 20.02 ± 3.60 | 8.70 ± 4.97 | 7.23 ± 5.33 |
| | Bleached | 4.67 ± 2.52 | 0.23 ± 0.23 | 21.05 ± 9.11 | 2.79 ± 2.12 |
| | Dead | 1.11 ± 0.74 | 0.45 ± 0.45 | 48.05 ± 11.88 | 47.66 ± 16.07 |

Mortality p = 0.0354) (Table 3). High levels of mortality (~ 48%) were later obeserved in heated tanks during the January assessments. Recruits in non-heated tanks had low levels of mortality (< 1%).

## Discussion

Results of this study reveal that *Pocillopora acuta* is able to recruit under projected levels of ocean warming and acidification; however, coral recruits are more susceptible to bleaching and mortality during the warmest months of the year. Additionally, ocean warming and acidification alone or in combination did not impact coral recruit size or abundance, but heating impacted subsequent health and survival. Although previous work has indicated that early life stages of many marine animals are more sensitive to environmental change compared to their adult counterparts [54–56], coral recruits of *P. acuta* appear to be a robust, weedy species that may be able to maintain high recruitment rates under pending climatic threats but will eventually succumb during the warmest months.

*Pocillopora acuta* in Hawaiʻi are able to produce planulae year-round, with optimum planulation levels between 26–27°C. The reproductive ability of *P. acuta* is curtailed when temperature shifts as little as 1°C from that range [57]. Once temperatures exceed 1°C above the optimum [26–27°C], the successful reproduction of *P. acuta* diminishes by up to one order of

magnitude [57]. Given these previous studies, we expected to see a difference in abundance across treatments between July and October, with the non-heated tanks having a greater abundance than the heated tanks. In contrast to these predictions, there was no significant difference in coral recruit abundance between July and October; however, there were significant declines in recruit abundance in heated treatments between October and January. This decline is due to high mortality levels in the recruits during the high temperatures in October.

Likewise, previous research investigating the impact of ocean acidification on newly settled coral recruits have reported strong effects including reductions in settlement (45–65% in *Acropora palmata*) [34], skeletal mass (20–60%) [39–41], overall mineral deposition [58], and deformed and porous skeletal structure [58]. Coral recruits exposed to acidified conditions (900 μatm) also had structurally compromised skeletons that were smaller (reduced diameter and height), more fragile (thinner basal plate, pitted or porous corallite walls), and asymmetric [58]. Given these previous studies, we expected to see negative impacts of acidification on recruits settling to low pH treatments. Again, in contrast to predictions derived from these previous studies, we detect no difference in abundance and/or size of coral recruits exposed to acidified conditions compared to control (present-day) conditions. Warming and acidification alone or in combination did not impact the recruitment (total or mean number) of *Pocillopora acuta*; however, high levels of mortality (48%) were observed among those recruits exposed to elevated temperatures treatments during summer temperatures. These high levels of bleaching associated mortality led to lower abundance numbers in the heated treatment tanks during the January 2018 assessment. Therefore, elevated temperature as opposed to acidification was the driver of observed mortality in the recruits.

Previous studies have highlighted that ocean warming and acidification may not necessarily have direct effects on corals, but rather on their habitat or settlement cues that can impact recruitment [59]. For example, the crustose coralline algae, *Neogoniolithon fosliei*, and its associated microbial communities cannot tolerate elevated temperatures of 32˚C. Significant changes in the microbiology (increase in *Bacteroidetes* and a reduction in *Alphaproteobacteria*), pigmentation, and photophysiology occur in *N. fosliei* at 32˚C which reduce induction of coral larval metamorphosis by 50% [60]. Further, field experiments at a $CO_2$ seeps in Papua New Guinea revealed that coral settlement rates, recruit and juvenile densities were best predicted by the presence of crustose coralline algae, as opposed to the direct effects of seawater $CO_2$ [59]. We do not observe such an impact on the recruitment of *P. acuta* in these experiments, but other coral species may be comprised if settlement cues are impacted. Additionally, more susceptible adult brood stocks may have impaired reproduction and therefore recruitment levels due to thermal stress, which could lead to a switch in dominant taxa. Recruitment driven changes in coral dominance have also been documented on both the Great Barrier Reef (GBR) and in Moʻorea. On the GBR, following the 2016–2017 bleaching event, larval recruitment declined by 89% in 2018 and for the first time brooding pocilloporids replaced acroporids as the dominant taxon in the recruitment pool [61]. Likewise, predation by a 2006–2010 outbreak of crown-of-thorns sea stars reduced coral cover by ~90% around the island of Moʻorea, followed by a 2010 cyclone which removed most of the dead coral skeletons [62]. Unusually high coral recruitment in 2011 and 2012 was dominated by pocilloporids which appear to be the early successional stage and the relationship between recruit density and coral cover of pocilloporids and acroporids likely mediates temporal shifts in taxonomic composition of coral communities [63].

Recovery and persistence of coral reef ecosystems is dependent on the ability of recruitment levels to keep pace with adult population mortality [42, 64]. However previous work suggests that the early life stages of many marine animals are more sensitive to environmental change compared to their adult counterparts [54–56]. The results of this study revealed that the weedy

species, *Pocillopora acuta*, was able to recruit under projected levels of ocean warming and acidification but elevated temperatures experienced during the warmest months resulted in high levels of bleaching and mortality compared to present-day (control) treatments. Overall results were most similar by temperature treatment regardless of pH level (Fig 5). Interestingly, coral recruits exposed to dual stress continued to increase in mean size despite these high levels of bleaching whereas corals exposed to heat alone had restricted growth. These results suggest that the combined effects of moderate levels of acidification and warming appear to offset one another, because combined stressor treatments that include acidification migitate the adverse effects seen in single stressor treatments where corals deal with lethal temperatures alone.

## Supporting information

**S1 Fig. Photograph inside of experimental tank showing adult colonies and coral recruits.** (TIFF)

**S2 Fig. Diurnal fluctuations in temperature (a.) and pH$_T$ (b.) values in experimental treatments of control (AT ACO2; blue solid line), acidified (AT HCO2; black dotted line), heated (HT ACO2; red solid line), and heated acidified (HT HCO2; purple dotted line).** Diel sampling occurred quarterly throughout experiment. Data shown here is from 27–28 June 2017. (TIFF)

## Acknowledgments

This work is dedicated to Dr. Paul Jokiel, a pioneer and visionary in the field of coral reef research and founder of the Coral Reef Ecology Laboratory, where this research was conducted. We would also like to thank two anonymous reviewers who contributed excellent suggestions and timely reviews of our work.

This publication is referenced as the University of Hawai'i's School of Ocean and Earth Sciences (SOEST) contribution number 10890 and Hawai'i Institute of Marine Biology (HIMB) contribution number 1784.

## Author Contributions

**Conceptualization:** Keisha D. Bahr, Christopher P. Jury.

**Data curation:** Keisha D. Bahr, Tiana Tran.

**Formal analysis:** Keisha D. Bahr, Tiana Tran.

**Funding acquisition:** Christopher P. Jury, Robert J. Toonen.

**Investigation:** Keisha D. Bahr, Tiana Tran, Christopher P. Jury.

**Methodology:** Keisha D. Bahr, Christopher P. Jury.

**Project administration:** Robert J. Toonen.

**Resources:** Robert J. Toonen.

**Supervision:** Keisha D. Bahr, Christopher P. Jury, Robert J. Toonen.

**Validation:** Keisha D. Bahr.

**Visualization:** Keisha D. Bahr.

**Writing – original draft:** Keisha D. Bahr, Tiana Tran.

**Writing – review & editing:** Keisha D. Bahr, Christopher P. Jury, Robert J. Toonen.

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
