## [Decision Letter · Decision Letter 0]

3 Nov 2019

PONE-D-19-26999

Abundance, size, and survival of recruits of the reef coral Pocillopora acuta under ocean warming and acidification

PLOS ONE

Dear Dr. Bahr,

Thank you for submitting your manuscript to PLOS ONE. After careful consideration, we feel that it has merit but does not fully meet PLOS ONE’s publication criteria as it currently stands. Therefore, we invite you to submit a revised version of the manuscript that addresses the points raised during the review process.

Both reviewers raised several questions about ms structure. Major criticisms concern literature update and methodological detail. The authors must take these critics into full account.

      We    would appreciate receiving your revised manuscript by Dec 18 2019 11:59PM. To enhance the reproducibility of your results, we recommend that if applicable you deposit your laboratory protocols in protocols.io, where a protocol can be assigned its own identifier (DOI) such that it can be cited independently in the future. For instructions see: http://journals.plos.org/plosone/s/submission-guidelines#loc-laboratory-protocols

We look forward to receiving your revised manuscript.

Kind regards,

Carlo Nike Bianchi

Academic Editor

PLOS ONE

Journal Requirements:

2. Please ensure that you refer to Figure 4 in your text as, if accepted, production will need this reference to link the reader to the figure.

Reviewers' comments:

Reviewer's Responses to Questions

**Comments to the Author**

1. Is the manuscript technically sound, and do the data support the conclusions?

Reviewer #1: Yes

Reviewer #2: Partly

2. Has the statistical analysis been performed appropriately and rigorously? 

Reviewer #1: Yes

Reviewer #2: Yes

3. Have the authors made all data underlying the findings in their manuscript fully available?

Reviewer #1: Yes

Reviewer #2: Yes

4. Is the manuscript presented in an intelligible fashion and written in standard English?

Reviewer #1: Yes

Reviewer #2: Yes

5. Review Comments to the Author

Reviewer #1: Manuscript PONE-D-19-26999

The authors present a study that aim to quantify the abundance, size, and survival of the coral Pocillopora acuta that recruited inside experimental mesocosms under current and projected condition of ocean warming and acidification.

Early life stages of corals are more sensitive to environmental change compared to their adult counterparts so the present manuscript face a relevant topic for the coral reefs ecology.

I think the ms is worthy to be accepted for the publication on PONE after some major and minor revisions as follow.

In my opinion the Discussion paragraph, need to be rearrange to better underline ms results. Generally, your results have been little debated while most of the text regards statements of other authors.

The initial sentence (line 254-261) is redundant and more suitable for the Introduction. You should integrate it where you speak about the importance of the coral recruitment to predict future trajectories of coral reef ecosystems (line 57-64).

Sometimes in the text there is no space between two consecutive words, check the document (line 42).

(Line 50-52) add this sentence or similar and references: “although, field studies revealed that even abiotic factors in the marine environment may interfere with the effect of ocean acidification on corals (Oprandi et al., 2019; Oporto-Guerrero et al., 2018; Januar et al., 2017) and biological feedbacks complicate such predictions (24)”.

Oprandi, A., Montefalcone, M., Morri, C., Benelli, F., & Bianchi, C. N. (2019). Water circulation, and not ocean acidification, affects coral recruitment and survival at shallow hydrothermal vents. Estuarine, Coastal and Shelf Science, 217, 158-164.

Oporto-Guerrero, T., Reyes-Bonilla, H., & Ladah, L. B. (2018). Presence of the reef-building coral, Porites panamensis, in a shallow hydrothermal field in the Gulf of California. Marine Biodiversity, 48(1), 703-708.

Januar, H. I., Zamani, N. P., Soedarma, D., Chasanah, E., & Wright, A. D. (2017). Tropical coral reef coral patterns in Indonesian shallow water areas close to underwater volcanic vents at Minahasa Seashore, and Mahengetang and Gunung Api Islands. Marine ecology, 38(2), e12415.

Reviewer #2: Bahr et al reported the effect of warming and acidification on the recruits’ survival of the resistant coral Pocillopora sp. (recognised as P. acuta based on its morphology only) during a long-term experiment in aquaria.

This study could merit to be published likely as a note, therefore reducing citations, figures and likely some text, because although not innovative, it improves knowledge on the effect of environmental change on corals’ survival, specifically on their early life stage which it might be the major bottleneck.

I have a number of questions and points the Authors need to clarify. My first, natural question to the Authors is: do you really planned this experiment or is it just an opportunistic way to publish a side paper from one main project? 40 tanks were used during a 22-months study to study recruits’ survival only of one brooder coral only! Settlement was observed on the tanks wall (no supports for settlement were provided)! The tank become the unit to be compared between treatments! There is no description of where the settlement happen (top, bottom, other supports, plastic, PVC, pump?). No description of the aquaria conditions (presence of CCA on the walls? Nude, clean aquaria?). How many colonies per tank? Their size? Did the Authors cleaned the aquaria? What about algae effect on the coral survival rate? How the Authors observed the recruits pigmentation? Visually I read, but in which way? Maybe using a camera underwater? Did the Authors ‘dive’ into the 70-L tanks? Is this method enough precise? Although the experiment started in March 2016 only recruits from June 2017 were reported. Why? These are the most prominent questions I had reading this ms.

Other points.

Because the difficult to distinguish among the different Pocillopora species, which could be found in sympatry, it should be critical to identify the colonies used in the experiment using classic and available mitochondrial markers.

The literature about OA is old!! Langdon, Kleypas, Gattuso, Erez, Smith made the story of OA, but this science has deeply changed and the story has improved during the last ten years. Please, cite one, two of these papers and report more recent ones. Specifically, the Authors forgot to mention (and to discuss!!) studies on the effect of OW and OA on the Pocillopora species. Because Pocillopora has become a model to study early life stages, some studies have been already published so far. Please, read at least Putnam and the last study by Bellworthy!

I have a concern about the experimental conditions and the treatments. One measurement made at midday per week of the environmental parameters and averaged per month (and during the three study periods only) is not enough to describe the experimental environment. Any chance to have some more data?

Finally, this study does not assess OA because pH is too close to ambient. Firstly, I was surprised to read pH 8.21 for the Ambient treatment which is quite high (i.e., pCO2 218 is a preindustrial value!). What’s more pH is in total scale value that is ca 0.1 less than measurements made in NBS scale. But more relevant to address my concern is that the High CO2 treatment have pHT 8.06-7.95 (654-478 pCO2). This is not what the Authors should test to assess the effect of OA and actually those values are in the normal range experienced even by front reef corals. Is there a mistake in the calculation? Honestly, I cannot accept it even if presented as “moderate acidification” in the last sentence of the ms.

6. PLOS authors have the option to publish the peer review history of their article (what does this mean?). If published, this will include your full peer review and any attached files.

Reviewer #1: No

Reviewer #2: No

---

## [Author Response · Author response to Decision Letter 0]

11 Dec 2019

We would like to thank the reviewers and the editor for these timely reviews. They have improved the manuscript tremendously. We have addressed all the concerns of the reviewers and included the responses here as well as tracked changes in the revised manuscript. 

Significant improvements have been made to the manuscript structure including a literature update and more detail for the methods. We have also included more data as requested by Reviewer 2. 

Detailed responses to each comment by the reviewer has been included in a "Response to Reviewer" document.

---

## [Decision Letter · Decision Letter 1]

9 Jan 2020

Abundance, size, and survival of recruits of the reef coral Pocillopora acuta under ocean warming and acidification

PONE-D-19-26999R1

Dear Dr. Bahr,

We are pleased to inform you that your manuscript has been judged scientifically suitable for publication and will be formally accepted for publication once it complies with all outstanding technical requirements.

With kind regards,

Carlo Nike Bianchi

Academic Editor

PLOS ONE

Additional Editor Comments (optional):

Reviewers' comments:

Reviewer's Responses to Questions

**Comments to the Author**

1. If the authors have adequately addressed your comments raised in a previous round of review and you feel that this manuscript is now acceptable for publication, you may indicate that here to bypass the “Comments to the Author” section, enter your conflict of interest statement in the “Confidential to Editor” section, and submit your "Accept" recommendation.

Reviewer #1: All comments have been addressed

Reviewer #2: All comments have been addressed

2. Is the manuscript technically sound, and do the data support the conclusions?

Reviewer #1: (No Response)

Reviewer #2: Yes

3. Has the statistical analysis been performed appropriately and rigorously? 

Reviewer #1: (No Response)

Reviewer #2: Yes

4. Have the authors made all data underlying the findings in their manuscript fully available?

Reviewer #1: (No Response)

Reviewer #2: Yes

5. Is the manuscript presented in an intelligible fashion and written in standard English?

Reviewer #1: (No Response)

Reviewer #2: Yes

6. Review Comments to the Author

Reviewer #1: (No Response)

Reviewer #2: (No Response)

7. PLOS authors have the option to publish the peer review history of their article (what does this mean?). If published, this will include your full peer review and any attached files.

Reviewer #1: No

Reviewer #2: Yes: Riccardo Rodolfo-Metalpa

---

## [Editor Report · Acceptance letter]

14 Jan 2020

PONE-D-19-26999R1 

Abundance, size, and survival of recruits of the reef coral Pocillopora acuta under ocean warming and acidification 

Dear Dr. Bahr:

I am pleased to inform you that your manuscript has been deemed suitable for publication in PLOS ONE. Congratulations! Your manuscript is now with our production department. 

With kind regards,

on behalf of

Dr. Carlo Nike Bianchi 

Academic Editor

PLOS ONE